# Surface Degradation of Thin-Layer Al/MgF_2_ Mirrors under Exposure to Powerful VUV Radiation

**DOI:** 10.3390/nano13212819

**Published:** 2023-10-24

**Authors:** Andrei Skriabin, Victor Telekh, Aleksei Pavlov, Daria Pasynkova, Anastasiya Podlosinskaya, Pavel Novikov, Valery Zhupanov, Dmitry Chesnokov, Viacheslav Senkov, Alexander Turyanskiy

**Affiliations:** 1Department of Power Engineering, Bauman Moscow State Technical University, Moscow 105005, Russia; telekh@bmstu.ru (V.T.); alekseipavlov@bmstu.ru (A.P.); ermolay0808@mail.ru (D.P.); apodlosinskaya@yandex.ru (A.P.); 2Luch Scientific Production Association, Podolsk 142103, Russia; novikovpa@sialuch.ru (P.N.); zhupanovvg@sialuch.ru (V.Z.); chesnokovda@sialuch.ru (D.C.); 3Lebedev Physical Institute of the Russian Academy of Sciences, Moscow 119991, Russia; senkov42@yandex.ru (V.S.); algeo-tour@yandex.ru (A.T.)

**Keywords:** VUV radiation, telescope mirrors, plasma emitters of radiation, surface degradation, light-induced ablation

## Abstract

Thin-layer Al/MgF_2_ coatings are currently used for extraterrestrial far-UV astronomy as the primary and secondary mirrors of telescopes (such as “Spektr-UF”). Successful Hubble far-UV measurements have been performed thanks to MgF_2_ on Al mirror coatings. Damage of such thin-layer coatings has been previously studied under exposure to high-energy electrons/protons fluxes and in low Earth orbit environments. Meanwhile, there is an interest to test the stability of such mirrors under the impact of extreme radiation fluxes from pulsed plasma thrusters as a simulation of emergency onboard situations and other applications. In the present studies, the high current and compressed plasma jets were generated by a laboratory plasma thruster prototype and operated as effective emitters of high brightness (with an integral overall wavelength radiation flux of >1 MW/cm^2^) and broadband radiation. The spectrum rearrangement and hard-photon cut-off at energy above *E_c_* were implemented by selection of a background gas in the discharge chamber. The discharges in air (*E_c_* ≈ 6 eV), argon (*E_c_* ≈ 15 eV) and neon (*E_c_* ≈ 21 eV) were studied. X-ray diffraction and reflectometry, electron and atomic force microscopy, and IR and visible spectroscopy were used for coating characterization and estimation of degradation degree. In the case of the discharges in air with photon energies of *E* < 6 eV, only individual nanocracks were found and property changes were negligible. In the case of inert gases, the energy fraction was ≈50% in the VUV range. As found for inert background gases, an emission of such hard photons with energies higher than the MgF_2_ band gap energy of ≈10.8 eV caused a drastic light-induced ablation and degradation of the irradiated coatings. The upward trend of degradation with an increasing of the maximum photon energies was detected. The obtained data on the surface destruction are useful for the design of methods for coating stability tests and an understanding of the consequences of emergencies onboard space research stations.

## 1. Introduction

Thin multilayer coatings are the basis of a number of optical elements such as selective mirrors and interference filters [1,2]. Thin Al and other protected metal coatings (such as Ag, Be, and Au) [3,4,5] are used for extraterrestrial far-UV astronomy in the structure of primary and secondary telescope mirrors (such as “Hubble” [6] and “Spektr-UF” [7]). These space observatories were designed for studying astronomical objects in a wide spectral range, involving far- and near-UV radiation (from the Lyman-alpha Ly-α line at ≈121.6 nm to the atmosphere cutoff wavelength at ≈290 nm) [8]. In this spectral range, the Al mirrors with a protective MgF_2_ upper film demonstrated a reflectance as high as ≈90% at the Ly-α line [9]. At the same time, an elimination or cracking of the MgF_2_ layer under open-space or terrestrial conditions could drastically decrease the mirror quality due to the formation of an alumina layer on the reflective Al film. Even an emergence of thin Al_2_O_3_ layers (with a thickness of ≈3.5–5.0 nm) suppressed a high reflectivity of photons with energies above 9 eV, which led to mirror failure [6]. Clearly, the deposition procedures must be implemented under high-vacuum conditions using electron beam/heat evaporation or ion beam sputtering [10,11]. As reported in [10], the presence of water vapor during mirror preparation led to a significant decrease in reflectivity (up to ≈15% at the Ly-α line). Based on grazing X-ray reflectometry and UV spectrophotometry, post-treatment heating with annealing at 250 °C was approved for enhancing spectral reflectivity, which was explained by an increase in MgF_2_ density and a decrease in Al and MgF_2_ surface roughness [9]. Such transformations effectively blocked oxygen diffusion from the environment to the inner Al layer through pores and defects in the upper protective MgF_2_ coatings.

The prediction of mirror stability in a space environment shows the necessity for the design of stability tests for different telescope mirrors under the influence of damage factors such as neutral, electron/proton, and even He^+^ fluxes; UV irradiation; chemical compounds; and thermal cycling [3,12]. As found in [13], a maximum degradation of protected Ag mirrors was detected under complex exposure to solar-equivalent UV, 10 keV electrons (1.4 × 10^18^ e^−^/cm^2^), and 5 keV protons (1.6 × 10^17^ p^+^/cm^2^) after 1436 h. The reflectivity deviation Δ*R* between the exposed and unexposed samples achieved up to Δ*R* ≈ 30% in the UV range. Thermal cycling from –80 to +35 °C (30 cycles) also led to optical degradation and reduced reflectivity up to Δ*R* ≈ 10% [14]. But another study [15] reported a negligible degradation of Al/MgF_2_ mirrors at 121.6 nm when irradiated with 1 MeV electrons and 5 MeV protons.

The employment of UV mirrors onboard extraterrestrial telescopes requires an evaluation of their stability under the impact of plasma jets and radiation fluxes to simulate the consequences of emergencies. As an example, CubeSat miniaturized satellites [16] with pulsed plasma thrusters [17] could be used as prospective and inexpensive space observatories, which can serve the needs of universities and small research teams. Such plasma devices ensure high radiation loads (up to *q_s_* ≈ 1 MW/cm^2^) on irradiated surfaces and the generation of broadband VUV radiation [18], which presents these devices as unequal laboratory tools for degradation tests and the material sciences. The radiation sources, based on a coaxial magnetic plasma compressor (MPC) [19], can obtain ultra-high brightness plasma temperatures of ≥40 kK [20]. Such extreme radiation loads induce evaporation of the coating and the emergence of heat stresses and local defects; they stimulate phase and chemical transformations and other changes in irradiated coatings [21,22]. So, the goals of the present study were an experimental estimation of Al/MgF_2_ mirror stability and an understanding of the degradation mechanisms under exposure to powerful and ultra-high-brightness VUV radiation on the surfaces located in the near-field zone of the plasma thruster channel.

## 2. Materials and Methods

### 2.1. Plasma Radiation Emitter

The flow out of the MPC plasma jet interacts with a background gas and thermalizes with the formation of strong shockwaves and powerful broadband radiation fluxes. The chemical composition of the background gas rearranges the discharge spectrum by “cutting off” hard photons at energies higher than *E_c_* [23]. For inert gases, the *E_c_* values are determined using the ionization energy of the gas (e.g., *E_c_* ≈ 21 eV for neon and *E_c_* ≈ 15 eV for argon). In the case of oxygen-containing media, the energy is “cut off” at the Schumann–Runge bands (*E_c_* ≈ 6 eV) [24]. The MPC-based emitters have a high efficiency of ≈0.3 [25] and they are suitable as a laboratory tool for experimental studies of material stability under extreme radiation loads with minimal mechanical impacts from plasma. A typical discharge time is about ≈40–50 μs [21].

The setup scheme is presented in Figure 1. A low-inductance capacitor (1) (Rustechgroup Ltd., Moscow, Russia) with a capacitance of *C* = 18 μF was charged up to *W* ≈ 3.6 kJ with a STEN-20 charger (2) (Emission Electronics Ltd., Tomsk, Russia). An input of a stored energy was implemented with a controlled pulsed thyratron (3) (Pulsed systems Ltd., Ryazan, Russia) to the coaxial MPC (4) with AISI 321 steel electrodes (with inner and external diameters of 6 and 34 mm, respectively) and an ablative caprolon dielectric insert installed horizontally in a grounded vacuum discharge chamber (see Figure 2a). High current discharges caused the light-induced ablation of the caprolon insert and the formation of a high-temperature zone (plasma focus) (5), which emitted high-energy photons to the sample (6). Recording of current dynamics was fulfilled with a Pearson current monitor 110 (Pearson Electronics, Palo Alto, CA, USA) (7) and Tektronix 2024b oscilloscope (Tektronix, Beaverton, OR, USA) (8). Typical values of maximum discharge currents were *I* ≈ 100–150 kA during the discharge time of ≈30–40 μs, as shown in Figure 1. Single exposures in air, argon (99.993%) and neon (99.999%) were studied at the background gas pressure of 200 torr. The scheme of Al/MgF_2_ mirror disposition in the chamber relative to the MPC channel is presented in Figure 2a, whereas the dimensions of electrodes are shown in Figure 2b. The irradiated samples were located collinear to the MPC axis at a distance of *l_s_* = 47 mm. The half of the sample farthest from the MPC channel was covered by a copper screen for its conservation. In Figure 2c, the photo of the MPC discharge evidences that a plasma jet and radiation fluxes propagated away from the MPC and the mirrors were exposed to the UV/VUV radiation mainly without a mechanical impact. Thus, the mirror degradation was caused by the surface irradiation only.

### 2.2. Measuring of Discharge Energy Fluxes

A TDX10 thermal detector (Thorlabs Inc., Newton, MA, USA) was mounted on a blackened solid copper disk and installed inside an extended and evacuated tract at a distance of *l_d_* = 786 mm from the plasma focus position. This sensor with an active area *F_d_* of 1 cm^2^ allowed us to record the photons with wavelengths up to ≈20 μm. The detector was connected in parallel to a PM302E dual channel optical power and energy meter (Thorlabs Inc., Newton, MA, USA) and Hantek oscilloscope (Hantek Electronic Co., Ltd., Qingdao, China). A projection of the focus position onto the detector area center was achieved with a bellows hose, which allowed precise movement of the detector. The discharge energy fluxes on the irradiated surface *I_s_* were calculated as
*I_s_* = *W_d_*∙(*l_d_*/*l_s_*)^2^/*F_d_*. (1)

Here *W_d_* is the energy values recorded by the sensor. The heat flux density *q_s_* was estimated as *q_s_* = *I_s_*/*τ_ex_*, where *τ_ex_* is the exposure time.

### 2.3. Coating Deposition

The samples of Al/MgF_2_ mirrors were prepared on the sitall (an analogue of ZERODUR^®^) substrates (40 × 20 × 10 mm) by thermal and electron beam evaporation from molybdenum crucibles with an Integrity-100 facility. The initial average substrate roughness was about *S_a_* ≈ 0.77 nm. As calculated with [26], the reflectivity of uncovered sitall was ≈4.4–4.8% in a wavelength range of ≈365–2325 nm. Thicknesses of aluminum *δ_Al_* = 50 nm and magnesium fluoride *δ_MgF_*_2_ = 80 nm layers were controlled by a quartz sensor, and their dimensional specification is shown in Figure 1.

### 2.4. Coating Characterization

The features of inner structures (thicknesses, density) of the mirrors were studied with X-ray reflectometry (XRR) within a grazing angle range of ≈0.36–0.9° and their phase composition was investigated with X-ray diffraction (XRD) within a 2*θ* range of ≈10–75° utilizing a Bragg–Brentano *θ*–2*θ* geometry (Compleflex-5 X-ray reflectometer/diffractometer, CDP systems, Moscow, Russia; CuK_α_ radiation, 0.154 nm). Fitting of the coating parameters was fulfilled with the procedure in [27]. The recorded XRD spectra were interpreted with the ICDD PDF-2 database [28]. The surface topography was studied with atomic force microscopy (AFM) (NTEGRA PRIMA microscope, Zelenograd, Russia) at several locations (see below) as defined by ISO 4287 [29]. The scan field was 20 × 20 μm. Reflectance was measured with a Cary300 spectrometer within a wavelength range of 200–800 nm at a step of 1 nm. The coating surface was visualized with a Zeiss Ultra plus scanning electron microscope (SEM) (Carl Zeiss, Oberkochen, Germany).

## 3. Results

### 3.1. Inner Structure and Properties of As-Deposited Al/MgF_2_ Mirrors

The XRR spectrum of the as-deposited Al/MgF_2_ bilayer and its fitting curve are presented in Figure 3a. The weak nature of the reflectivity oscillations was caused by the similarity of layer and substrate densities. The XRR densities of MgF_2_ and Al layers were 2.8 and 2.67 g/cm^3^, respectively. These values were slightly less than for bulk ones (3.18 and 2.7 g/cm^3^ for MgF_2_ and Al samples, respectively). The recorded spectra demonstrated a reflectivity of ≈80% in the UV-A and visible ranges (a wavelength range of 350–600 nm). With wavelength decreasing and passing into the UV-B and UV-C ranges, reflectivity reduction was detected (up to ≈25% at 200 nm). The recorded spectrum is presented in Figure 4. AFM showed (see below) that the average surface roughness was about *S_a_* = 1.38 nm, the surface skewness was *S_sk_* = 0.70, and the coefficient of kurtosis was *S_ku_* = 1.35. In addition, individual elongate MgF_2_ crystallites with a maximum height of *S_p_* ≈ 16.78 nm were found.

The as-deposited XRD pattern contained a lot of X-ray reflexes (see Figure 5). Based on hexagonal *β*-eucryptite (PDF No. 70-1580), monoclinic *α*-spodumene (PDF No. 75-1091), and tetragonal *β*-spodumene (PDF No. 35-0797), the major XRD peaks of sitall were recorded. Moreover, an amorphous halo within a 2*θ* range of ≈10–40° was detected. The reflex at ≈38.5° was interpreted as cubic Al (PDF No. 04-0787), whereas MgF_2_ was not confidently found due to the substrate influence.

### 3.2. Surface Degradation of Irradiated Samples

We found that irradiation conditions significantly influenced the physical and chemical processes at the irradiated surface. In the case of irradiation by low-energy photons (*E* < 6 eV, discharges in air), no phase transformations were detected in the XRD spectrum, which was close to the as-deposited one (see Figure 5a).

The XRR spectrum of the irradiated coating was close to the curve (see Figure 3a) for the as-deposited mirror that evidenced a slight irradiation influence on the inner structure and the layer thicknesses. No macrodefect or damage on the surface was visible to the naked eye. The measurement locations for the irradiated and as-deposited reference surfaces are also presented in Figure 3a. The irradiation in air caused a slight decrease in reflectance. The reflectance deviation from the reference spectrum was about ≈1–4%, as shown in Figure 4b. A reason for this phenomenon is the surface degradation at micro/nanoscales. Only individual point defects were found due to the generation of debris from the MPC electrodes. Additionally, a slight modification of the coating surface was detected with AFM. The 3D, 2D, and roughness profiles are presented in Figure 6. The maximum and average roughness values were *S_p_* = 18.27 nm and *S_a_* = 1.72 nm, respectively (point 1). So, a slight increase in roughness was detected. The surface skewness was positive at *S_sk_* = 0.54 and less than for the as-deposited one (*S_sk_* = 0.70). The surface kurtosis *S_ku_* decreased from 1.35 to 0.44 but it was always less than 3.

In the case of the discharges in argon, the maximal photon energy was up to ≈15 eV. AFM images of the irradiated coating are presented in Figure 6. The degradation pattern depended on the sample position relative to the plasma source. Here and below, AFM image positions correspond to the points in Figure 4a. The area nearest to the MPC area (point 2) undergoes a significant degradation that is apparent by its conspicuous light-induced erosion at a distance of ≈20 mm. Less surface damage was detected at greater distances (point 3). The clear border between the reference and irradiated areas was visualized. In this case, non-uniform coating degradation did not allow us to use XRR for estimation of its ablation. In contradiction to the irradiation in air, a lot of cracks (with a depth of ≈100–120 μm, i.e., compatible with the total coating thickness of *δ_Σ_* = *δ_Al_* + *δ_MgF_*_2_ = 130 nm) were formed in the coatings. Moreover, local surface defects with a depth up to ≈170 nm were detected that evidenced at least local destruction of the sitall substrate. A typical thickness loss was about ≈30–40 nm. We detected a non-uniform crack distribution in the far area. The trend of a decrease in crack number was found when moving away from the MPC. Local round defects with a size of ≈50–100 μm were visualized in the coating due to the emission of molted debris from the electrode unit. The tracks of the molten drop drag were also visualized. The roughness also depended on the sample position. The *S_a_* values were *S_a_* = 19.44 nm at point 2 and *S_a_* = 14.96 nm at point 3. The maximum roughness increased up to *S_p_* = 217.72 nm at point 1 and *S_p_* = 107.13 nm. The surface skewness changed the sign: *S_sk_* = −0.89 at the area nearest to the MPC (point 2) and *S_sk_* = 0.41 at the far area (point 3). The coefficient kurtosis changed from *S_ku_* = 3.50 (at point 2) to *S_ku_* = 0.13 (at point 3). The data on spectral measuring at the indicated locations are presented in Figure 3b. As detected, the VUV exposure significantly deteriorated the reflectivity values due to the surface damage. The reflectance deviations from the reference spectrum were up to ≈19–37% (at point 2) and ≈12–16% (at point 3). So, the main degradation mechanisms were related to the cracks and point defect formation as well as the partial light ablation of the upper MgF_2_ layer. The main feature of the irradiated surface was the presence of the slight reflex of cubic Al (PDF No. 04-0787), which was detected in the coatings. Figure 5b also presents the differences between the reference XRD spectrum of the as-deposited Al/MgF_2_ mirror and that after the VUV exposure. The reflexes at ≈31.3°, 48.9°, and ≈57.4° can be interpreted as monoclinic α-spondumene (PDF No. 75-1091) whose concentration increased due to the intensive heat loads and stimulation of the phase transformations.

When irradiating with a photon energy up to ≈21 eV (discharge in neon), practically no traces of the coating were detected. Only the light-corroded substrate (with a number of small craters) and individual separate Al fragments were visible. The reflectivity practically corresponded to the uncovered sitall (≈4.4–4.8%). The 3D and 2D AFM profiles evidenced damage of the surface. The roughness increased even more and was about *S_a_* ≈ 179.07 nm. The maximum roughness was *S_p_* = 1017 nm. These and other parameters such as the *S_sk_* and *S_ku_* values were quite uniform along the irradiated surface. The surface skewness was *S_sk_* = 0.30 and the coefficient of kurtosis *S_ku_* = −0.86. Point defects due to the molten debris drag were also visualized. Only insignificant traces of the Al layer were detected. For interpretation of the phase transformations in the substrate, the difference pattern between the as-deposited coatings and those after exposure to neon is presented in Figure 5b. We note that the irradiation with photon energy of ≈15–20 eV caused the appearance of intensive XRD reflexes at ≈31.3°, 48.9°, and 57.4°, which also corresponded to monoclinic α-spondumene (PDF No. 75–1091). The larger number (in comparison with the irradiation in Ar) of intensive peaks evidenced a large degree of phase transformation in the irradiated substrate.

As presented in Figure 7, SEM images of the irradiated surfaces additionally allowed us to clarify the details of their degradation features at small scales. Nanocracks were visualized for the coatings irradiated in air. An emergence of such damage could be caused by a high-brightness irradiation with relatively low-energy photons (*E* < 6 eV) due to purely thermal processes. They can play a key role in the minor reflectivity decreasing in the near- and middle-UV ranges due to the formation of channels through which oxygen and water vapors can penetrate the Al layer and slightly oxidize it. In the case of exposure to higher-energy photons (*E* ≈ 15–20 eV), degradation was more pronounced and confirmed by AFM imaging.

A lot of craters, cracks, and evaporation traces were found in this case.

### 3.3. Energy Fluxes from Discharges

Radiation flux measurements allow us to estimate the conversion efficiency of stored energy in the capacitor $W_{c}=CU_{0}^{2}/2$ into a radiation energy $W_{r}=4\pi l_{d}^{2}W_{d}/F_{d}$ as *η = W_r_/W_c_*. In the studied cases, the *η* values were varied from ≈0.11 to ≈0.41 depending on the experimental conditions and the background gas. The smaller values of the conversion efficiency corresponded to discharges in air. This could be explained by energy losses per activation of the non-equilibrium chemical reactions between oxygen and C and N atoms. Calculated with Equation (1), the *I_s_* values were *I_s_* ≈ 720–1260 mJ/cm^2^ for air and *I_s_* ≈ 1860–5580 mJ/cm^2^ for the inert media. In the case of the typical values of *τ_ex_* ≈ 5 μs, the heat flux densities were *q_s_* ≈ 0.14–0.25 MW/cm^2^ for air and *q_s_* ≈ 0.37–1.12 MW/cm^2^ for the inert gases. Significant differences in the integral overall wavelength heat flux density for neon and argon were not detected in our studies with the used sensor.

## 4. Discussion

The performed experimental studies demonstrated the sufficient roles of the maximum photon energy and heat flux densities in the near-surface processes. The surface temperature *T_s_* under exposure to the VUV radiation can be estimated as [21]
(2)$$T_{s}=T_{0}+\bar{A}q_{s}\sqrt{\frac{\pi \tau_{ex}}{\lambda \rho C_{p}}}$$

Here, *T*_0_ = 300 K is the temperature of the opposite non-irradiated surface, *λ* = 146 W/(cm∙K) is the heat conductivity, *ρ* = 2.53 g/cm^3^ is the density, and *C_p_* = 0.8 J/(g∙K) is the heat capacity [30]. The thermal physical properties were accepted for ZERODUR^®^. Key roles of the Al/MgF_2_ coating in the degradation are maintenance of an integral overall wavelength absorptivity $\bar{A}$ (or reflectivity) and prevention of overheating.

In the case of discharges in air, the minimum photon wavelength is about *λ_c_* ≈ 200 nm (with the maximum energy of *E_c_* ≈ 6 eV). The integral absorptivity values were calculated with OpenFilters 1.1 open software [31] with the data on complex refractive indexes of the thin MgF_2_ and Al layers on the bulk ZERODUR^®^ substrates [26]. These values were quite low ($\bar{A}\leq 0.05$), and the surface temperature was also low (*T_s_* ≈ 460–590 K). Here, MgF_2_ dielectric was almost transparent in the visible and UV ranges while the Al layer reflected the incident radiation. Such heat loads did not lead to visible light-induced ablation. The emergence of the nanocracks and the insignificant decrease in reflectivity could be explained by the heat impact from the shockwaves on the surface and caused by Al oxidation through the formed defects. The emission of harder photons caused drastic changes to the degradation processes. The MPC discharges in argon were characterized by the hard photon parameters of *λ_c_* ≈ 80 nm and *E_c_* ≈ 15 eV, which are equivalent to the ionization energy. In this case, the photon energy was higher than the MgF_2_ band gap energy of ≈10.8 eV. So, the MgF_2_ layer effectively absorbed the incident VUV radiation and transformed it into the internal energy that led to its partial overheating and ablation. The reflectivity of the Al layer also decreased, whereas the absorptivity increased up to $\bar{A}\approx 0.08-0.1$, and the surface heating could be up to *T_s_* ≈ 980–1100 K that slightly exceeded the melting point of aluminum (960 K). The melting of the Al layers was also discovered with SEM imaging. In the case of the photon energies up to *E_c_* ≈ 21 eV (*λ_c_* ≈ 57 nm; discharges in neon), the absorptivity was drastically increased (up to $\bar{A}\geq 0.3$), and the surface overheating was *T_s_* > 2500 K. Full light-induced ablation of the mirrors and the upper substrate layers was detected. Such heat loads stimulated the forming of high and extended crystalline peaks (with a height of ≈1 μm) on the surface and the *β–α*-spondumene phase transformation [32].

The achieved experimental conditions were unique in terms of the photon energies, heat flux densities, and irradiated areas. The light-induced degradation was much more extreme than in other experimental studies [13], which were focused on the electron/proton exposure; thermal cycling; solar-equivalent UV impact; and the surface erosion with He^+^ and Ar^+^/UV treatment [12,33]. The studied exposure mode was characterized by radiation exposure of large areas unlike laser-induced ablation [34]. We believe that the presented experimental data are useful for an understanding of the degradation mechanisms, which could take place during space operations of miniaturized CubeSat satellites with pulsed ablative thrusters and UV telescopes onboard.

## 5. Conclusions

The dense and high-current discharges generated high-brightness and broadband VUV radiation with the coaxial MPC tool. This emitter demonstrated high efficiency of *η* ≈ 0.11–0.41 and integral overall wavelength energy fluxes of *I_s_* ≈ 1260–5580 mJ/cm^2^. The discharges in the background gases allowed us to implement spectrum rearrangement and control the maximum photon energies *E_c_*. In the case of the VUV radiation, the photon energies drastically influenced the thermal near-surface processes. For discharges in air with *E_c_* ≈ 6 eV, only minor changes were detected. But for discharges in inert gases with *E_c_* ≈ 10–20 eV, extensive light-induced ablation was detected due to the high surface temperatures (*T_s_* > 2500 K). Such irradiation also led to phase/chemical changes in the upper substrate layers due to non-equilibrium high-temperature loads.

## Figures and Tables

**Figure 1 nanomaterials-13-02819-f001:**
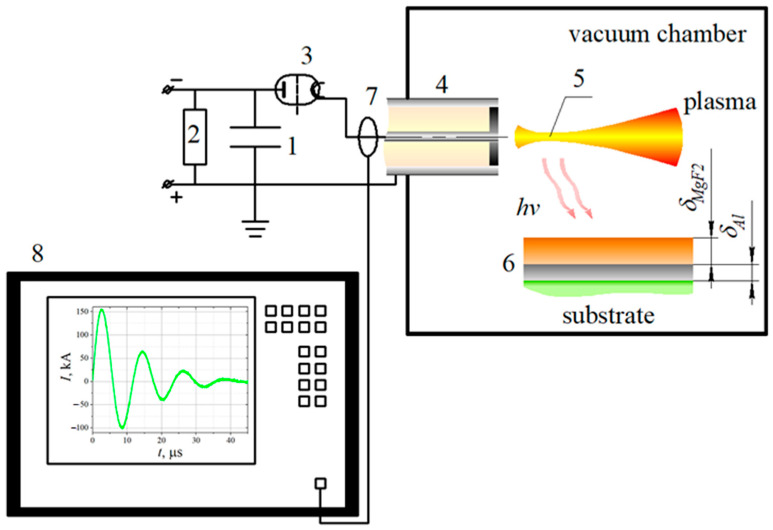
Scheme of experimental setup: 1—capacitor, 2—charger, 3—thyratron, 4—MPC, 5—plasma focus, 6—irradiated sample, 7—current monitor, 8—oscilloscope.

**Figure 2 nanomaterials-13-02819-f002:**
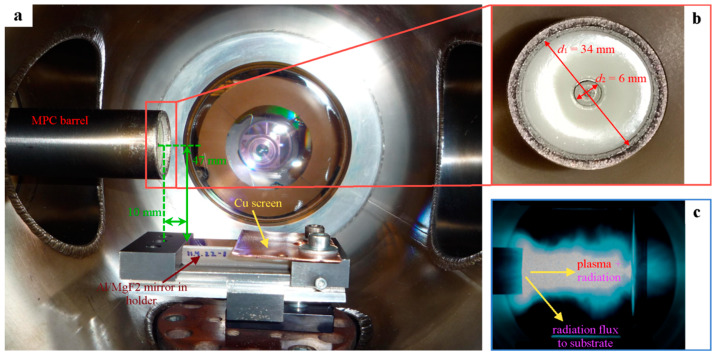
Scheme of Al/MgF_2_ mirror position in chamber (**a**), electrode dimensions, (**b**) and photo of discharge (**c**) in argon (stored energy of 3.6 kJ).

**Figure 3 nanomaterials-13-02819-f003:**
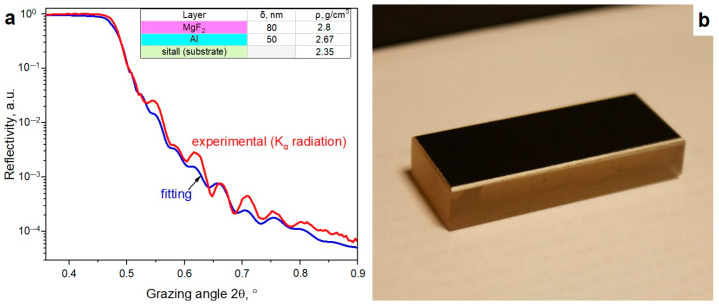
XRR spectrum with its fitting and inner structure (**a**) for prepared Al/MgF_2_ bilayer on substrate (**b**).

**Figure 4 nanomaterials-13-02819-f004:**
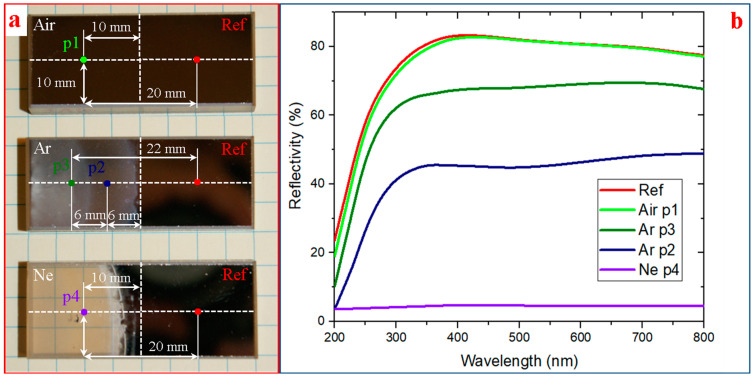
Photos of irradiated coatings (**a**) and their spectral reflectivity (**b**) in indicated points.

**Figure 5 nanomaterials-13-02819-f005:**
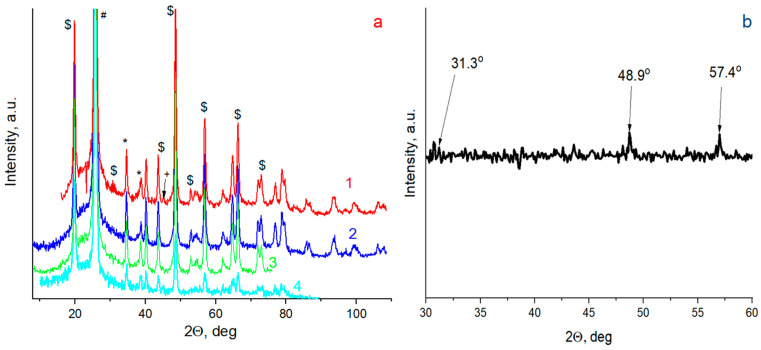
(**a**) XRD patterns of samples (1—argon, 2—neon, 3—air, 4—as-deposited): + Al (PDF No. 04-0787), # *β*-spodumene (PDF No. 35-0797), * *β*-eucryptite (PDF No. 70-1580), $ *α*-spodumene (PDF No. 75-1091); (**b**) differential XRD pattern of surface irradiated in neon.

**Figure 6 nanomaterials-13-02819-f006:**
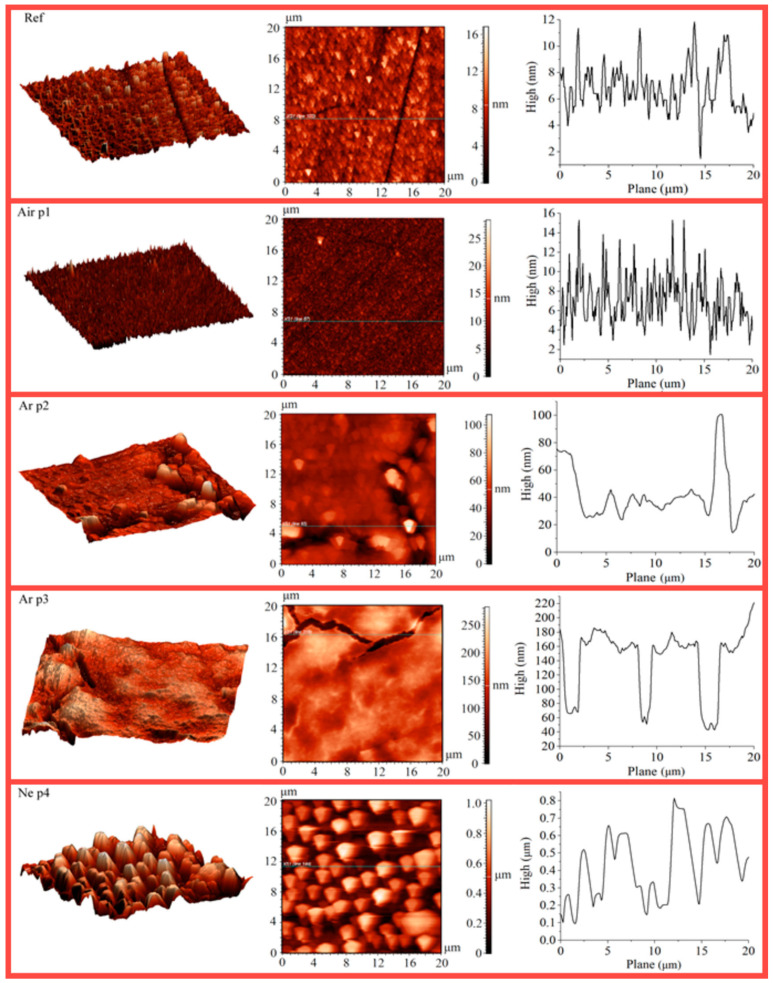
AFM images of sample surfaces at different points.

**Figure 7 nanomaterials-13-02819-f007:**
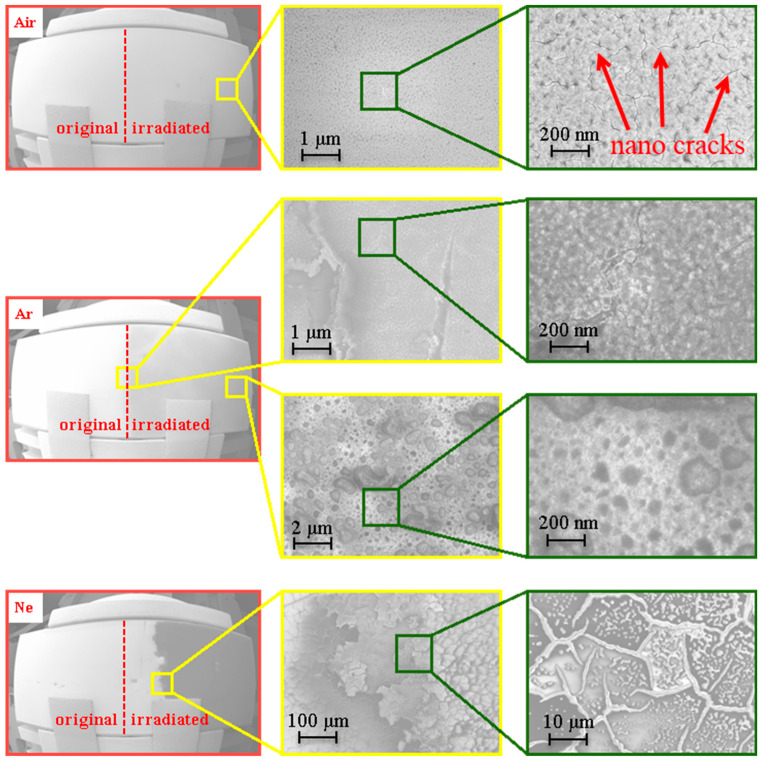
SEM images of sample surfaces at different points.

## Data Availability

The data presented in this study are available on request from the corresponding author.

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
