# Peer review of "Surface Degradation of Thin-Layer Al/MgF2 Mirrors under Exposure to Powerful VUV Radiation"

_nanomaterials, 2023, doi:10.3390/nano13212819_

Round 1

Reviewer 1 Report

In this paper the authors investigated the surface degradation of thin-layer Al/MgF2 mirrors under exposure of powerful VUV radiation. these kind of experiments are rather interesting to test and verify the stability of optical coatings to be used (in particular) in space applications.  The results reported in the manuscript demonstrated how high energy exposure can seriously damage the coating performances confirming previously reported studies.

Even if the technical contents are good and interesting, I think that the paper needs a significant revision before to be accepted for publication.

First of all, the abstract is highly confused. It should introduce and summarize the paper, while here it is not clear to topic and the goal of the study. I would say the same for the introduction. There is no real introduction of the experiments performed here and the use of a large number of very short english sentences makes the reading somehow "weird".

Also the material and methods should be improved. it is hard to understand the setup used and the different steps of the experiments. moreover , at line 133 the authors say "As found [35], a reflectivity was ≈4.4-4.8 % within a wavelength of ≈365-2325 nm" is this the reflectivity of the Al coating??? 4.4%??.

the results and discussion part is better and quite convincing. my only comment is:

- line 234 "Figure A1 (B) also presents" , what is Figure A1?

The authors should also check the references (there are some repetitions). Finally they can consider to mention another recent paper on similar experiments (Sci. rep. 11, 3429 (2021))

the english language is very confused. in particular in the first part of the manuscript (abstract, introduction and methods). I recommend a radical revision. (see my comments)

Author Response

Dear Colleague!

First of all, we would like to thank you for interest in the work.

Concerning you remarks we can note the following

  1. “First of all, the abstract is highly confused. It should introduce and summarize the paper, while here it is not clear to topic and the goal of the study. I would say the same for the introduction. There is no real introduction of the experiments performed here and the use of a large number of very short english sentences makes the reading somehow "weird" Also the material and methods should be improved. it is hard to understand the setup used and the different steps of the experiments.”.

We fixed Abstract, Introduction and Methods sections according to your recommendation (see Manuscript)

  1. “at line 133 the authors say "As found [35], a reflectivity was ≈4.4-4.8 % within a wavelength of ≈365-2325 nm" is this the reflectivity of the Al coating??? 4.4%??.”

No. It is the reflectivity of the uncovered substrate. The text correction has been made.

  1. “- line 234 "Figure A1 (B) also presents" , what is Figure A1?”

Figure A1 was presented in Supplementary section. It has been moved to Results.

  1. “The authors should also check the references (there are some repetitions). Finally they can consider to mention another recent paper on similar experiments (Sci. rep. 11, 3429 (2021))”

We have fixed it.

Reviewer 2 Report

Comments and Suggestions for Authors 

The authors studied compressed plasma jets generated by pulsed plasma discharges, which are effective emitters of high brightness and broadband radiation. The discharge energy is ≈50 % within a wavelength range of < 200 nm for discharges in inert gases. The presence of hard photons causes a drastic light-induced ablation and degradation of irradiated surfaces. Al/MgF2 bilayer is used for the task of extraterrestrial far UV astronomy in the design of the primary and secondary telescope mirrors. The authors tested the mirror stability under the impact of radiation fluxes from pulsed plasma thrusters. Only individual nano-cracks were found in the case of λ>200 nm (discharge in air). The emission of hard photons caused the catastrophic coating ablation, optical degradation, and roughness increase. Measuring integral radiation fluxes and estimating the surface temperature testified to the thermal mechanisms of the degradation.

This research is interesting for optical coatings for mirrors used in space. However, some of the issues in the manuscript should be illustrated.

Line 135:  What is “δAl=50 nm and δMgF2=80 nm”?  What is the thickness of the Al and MgF2?  The thicknesses should be illustrated. The two thicknesses of the films may affect the real reflectivity of Al/MgF2 in irradiation degradation.  

In Figure 4: The 3D AFM image in the left image does not show high dimensions.

Line 165: What is “ (see below)” ?  In the study, the XRD peaks can be produced in the powerful VUV radiation. However, the XRD patterns are shown in Figure A1.  And, the major peaks are not pointed out in the figure. Figure A1 should be written in this section as it is easy to read and understand.

Line 306: “The reflectivity of the Al layer also decreased. This led to the absorptivity increasing up to ?Ì… ≈ 0.08 − 0.1,………..”  A decrease in reflectivity does not lead to an increase in absorption because the sample has transmittance. That can be seen in P4 of Figure 3. And, how to measure the sample absorption in the study. 

Extensive editing of the English language required

Author Response

Dear Colleague!

First of all, we would like to thank you for interest in the work.

Concerning you remarks we can note the following

  1. “Line 135:  What is “δAl=50 nm and δMgF2=80 nm”?  What is the thickness of the Al and MgF2?  The thicknesses should be illustrated. The two thicknesses of the films may affect the real reflectivity of Al/MgF2 in irradiation degradation. ”

We have added the dimensional specification in Figure 1.

  1. “In Figure 4: The 3D AFM image in the left image does not show high dimensions.”

The high dimensions of the surface profile are better visible in the right image section where the profile dimensions along a test pathway were presented.  

  1. “Line 165: What is “ (see below)” ?  In the study, the XRD peaks can be produced in the powerful VUV radiation. However, the XRD patterns are shown in Figure A1.  And, the major peaks are not pointed out in the figure. Figure A1 should be written in this section as it is easy to read and understand.”

Ok. We have added the major peaks and the XRD patterns in Figure 5 (see Results section).

  1. “The reflectivity of the Al layer also decreased. This led to the absorptivity increasing up to ?Ì… ≈ 0.08 − 0.1,………..”  A decrease in reflectivity does not lead to an increase in absorption because the sample has transmittance. That can be seen in P4 of Figure 3. And, how to measure the sample absorption in the study. 

For VUV irradiation of materials, an attenuation length la is extremely short. As found with the Berkley Lab online resource (https://henke.lbl.gov/optical_constants/atten2.html), the la values are less than 0.05 microns (for MgF2) and less than 0.8 microns (for Al) for the photons with wavelengths of 10-90 nm. In the case of sitalle substrates, the radiation absorption takes place in a thin top layer. So, the transmittance is negligible for hard photons (≈10-100 nm). As mentioned in Manuscript, the absorption was numerically estimated with the standard open source software and known data on the refractivity index and the mirror structure but not measured. This evaluation is justified to clarify the degradation mechanism.    

Round 2

Reviewer 1 Report

The authors improved the manuscript a lot.

Now it is ok for publication 

Reviewer 2 Report

The manuscript has been revised as required.